# A Study on High and Low Temperature Rheological Properties and Oil Corrosion Resistance of Epoxy Resin/SBS Composite Modified Bitumen

**DOI:** 10.3390/polym15010104

**Published:** 2022-12-27

**Authors:** Zhuo Xue, Wenyuan Xu

**Affiliations:** School of Civil Engineering, Northeast Forestry University, Harbin 150040, China

**Keywords:** epoxy resin, oil corrosion, temperature sweep, multiple stress creep test, high and low temperature performance, fluorescence microscopy

## Abstract

In order to investigate the high and low temperature rheological properties and fuel corrosion resistance of epoxy resin on SBS modified asphalt, epoxy resin/SBS composite modified asphalt (ER/SBS) was prepared by high-speed shear. Moreover, composite modified bitumen with different proportions were designed based on the uniform design method and the basic performance index test was performed. The optimal composite mixing ratio of the ER and SBS modifier in composite modified asphalt (2.3% and 3.8%, respectively) was determined. Temperature scanning and a multiple stress creep test (MSCR) on ER/SBS composite modified asphalt with different ER content before and after oil corrosion was carried out using a dynamic shear rheometer (DSR). In addition, the high temperature rheological properties of different ER contents and composite modified asphalt after oil corrosion were evaluated by combing DSR measurements with the test data. The creep stiffness (S) and creep rate (m) indexes were obtained by a bending rheometer (BBR), and the effect of ER on the low-temperature rheological properties of SBS modified bitumen was investigated. The influence of the modifier incorporation on the micromorphology of asphalt and the change of micromorphology of asphalt after oil corrosion were analyzed by fluorescence microscopy. The test results show that the incorporation of 2.3% ER and 3.8% SBS can effectively improve the high temperature performance of SBS modified asphalt under the premise of cost saving. Moreover, the composite modified asphalt doped with ER can effectively improve the resistance of SBS modified asphalt to fuel corrosion at high temperatures, and the greatest improvement in the oil corrosion resistance of composite modified asphalt is observed at the ER content of 2.3%.

## 1. Introduction

With the development of the transportation industry, vehicle oil leakages are common. At present, the majority of roads are mostly asphalt pavements. Asphalt is a complex polymer mixed material made up mainly of alkanes, naphthenes, aromatic hydrocarbons, and non-metallic derivatives [1]. Bitumen is a hydrocarbon product extracted from petroleum, whose chemical composition is similar to organic solvents such as gasoline and diesel [2]. Therefore, it is easily dissolved by organic solvents such as gasoline and diesel. Consequently, asphalt pavements are eroded by oil, which is further accelerated under the action of high temperatures, rain, and vehicle load in summer; oil corrosion damage to road surfaces develops from the initial “oil corrosion belt” to the “pothole belt”. This has a great impact on the bearing capacity, durability, comfort, and driving safety of the road surface [3]. Li et al. investigated the oil corrosion mechanism of asphalt and proposed the concept of oil corrosion. Th authors evaluated the oil corrosion effect of diesel on 70# asphalt and PG-82 modified asphalt at different temperatures and determined that the oil corrosion resistance of modified asphalt is stronger than that of matrix asphalt according to the oil corrosion degree [4]. Li et al. tested the mechanical properties (e.g., using a modified turbidity test) on different asphalt mixtures to evaluate the fuel corrosion resistance of asphalt mixtures [5]. These tests did not damage the test pieces, saving time and generating economic benefits, while the tested sample can be used for further mechanical testing. Xu et al. evaluated the diesel corrosion resistance of TLA modified asphalt using the Marshall stability test, a freeze–thaw splitting test, and a rutting test, and found that the incorporation of TLA could improve the high temperature stability and water stability of asphalt mixture after oil etching [6]. Du et al. modified the asphalt slurry bonding strength (BBS) test and developed a new asphalt fuel corrosion resistance test method, which has the advantages of a simple operation, high precision, and good discrimination. In order to overcome the problem of oil corrosion on asphalt pavements, the majority of research promotes the development of oil-resistant modified asphalt by modifying the matrix asphalt [7]. Ma et al. evaluated the oil corrosion resistance of five types of asphalt horseshoe crushed stone (SMA) mixtures. The results show that the asphalt mixture with the oil corrosion inhibitor can maintain excellent mechanical and road properties after oil corrosion damage compared to the SBS and matrix asphalt mixture [8]. Gao et al. evaluated the fuel corrosion resistance of silicone resin as a fog sealing layer material by using contact angles, the Cantabro test, a water stability test, a wheel tracking test, and a three-point bending test. The results show that silicone resin can effectively improve the adhesion, strength, water stability, and high and low temperature performance of asphalt mixture before and after fuel oil erosion, and the improving effect on asphalt mixture after oil erosion is better [9].

In modern highway constructions, styrene-butadiene-styrene block copolymer (SBS) is often used as a modifier in modified asphalt concrete, and with the increase of road traffic pavement oil corrosion has become a potential source of early damage to asphalt pavements. Fuel leakages can induce corrosion damage to the road surfaces of SBS modified asphalt.

It is found that the resin corrosion resistance ability of different systems is different, and in road engineering epoxy resin as a crosslinking curing resin has good corrosion resistance and excellent mechanical properties [10,11]. Epoxy resin is often used as a modifier in asphalt concrete. Epoxy bitumen (EA), one of the most typical thermosetting polymer modified bitumens, is mainly composed of matrix bitumen, epoxy resin (ER), a curing agent, and other additives. As a bridge deck pavement material, it has the advantages of high strength, good fatigue resistance, and excellent high temperature and water stability, as well as corrosion resistance. Thermosetting three-dimensional network polymers are formed by the reaction of ER with bitumen and the curing agent in epoxy bitumen [12,13]. The presence of a curing network provides the EA with good adhesion, thermal stability, and anti-aging properties [14,15,16]. It can resist, to a large extent, the problem of road corrosion caused by fuel leakage during vehicle driving. However, due to its insufficient toughness at low temperatures, the poor compatibility between ER and asphalt, and its high price, it is not widely used on asphalt pavements [17,18]. The SBS modified asphalt mixture has a good low-temperature crack resistance, and the incorporation of SBS can improve the low-temperature fracture toughness of EA [19,20]. In addition, the preparation of modified epoxy bitumen by SBS can enhance the compatibility of ER with matrix bitumen and the flexibility of cross-linked epoxy resin networks. Xu et al. investigated the mechanism of the SBS capacitance for ER through microscopic experiments, including fluorescence microscopy, microcomputed tomography, and gel chromatography [21]. It must be considered that ER is often used in bridge deck pavement projects and the cost of using a large amount is high [22]. The current paper combines low-content ER and SBS to compose the matrix asphalt. We then explore the optimal formula and dosage combination of ER and SBS modifiers through indoor tests and evaluate the performance and oil corrosion resistance of ER/SBS composite modified asphalt. This work provides theoretical support for low-content EA applications in asphalt pavement susceptible to fuel corrosion.

## 2. Materials and Methods

### 2.1. Test Materials

The test materials were matrix bitumen (SK-90# matrix bitumen, Xingtai Baitong Asphalt Sales Co., Ltd., Xingtai, China), SBSYH-792E (Brand: KRATON, Model: D-KX410CS, Jinan Shanhai Chemical Technology Co., Ltd., Jinan, China) modifier, and E-44 epoxy resin (Brand: AILIKE, Model: E-44(6101), Dongguan Ailike New Materials Co., Ltd., Dongguan, China). Table 1, Table 2 and Table 3 report the technical indicators of SK-90 matrix asphalt, SBSYH-792E, and E-44 epoxy resin.

### 2.2. Preparation of Epoxy Resin/SBS Composite Modified Asphalt

The ER/SBS composite modified asphalt was prepared as follows: (1) add a fixed amount of SBS modifier to the heated and molten matrix asphalt; (2) prepare the SBS modified asphalt with the help of a mixer and high-speed shear mechanism; (3) place the SBS modified asphalt in the oven for swelling and development; (4) mix and stir the epoxy resin and curing agent at 60 °C at a ratio of 1:1 to prepare the epoxy system; (5) high-speed shear the prepared epoxy system and SBS modified asphalt through a shear; and (6) solidify the ER. Figure 1 presents the preparation process of the SBS composite modified bitumen. 

### 2.3. Test Methods

#### 2.3.1. Basic Index Test of Composite Modified Asphalt Based on the Uniform Design Method

The uniform design method was used to design the combined dosage of the ER and SBS modifiers based on the Test Regulations for Asphalt and Asphalt Mixtures for Highway Engineering (JTG E20—2011) [23]. The test methods and experimental procedures tested the penetration degree (25 °C), softening point (R&B), ductility (5 °C), and cloth viscosity (135 °C) of the ER/SBS composite modified asphalt.

#### 2.3.2. Determining the Optimal Composite Content of Composite Modified Asphalt

Laboratory testing of the ER/SBS composite modified asphalt penetration, ductility, softening point, and cloth viscosity were combined with regression analysis (SPSS, version 26.0, IBM, Chicago, IL, USA) to determine the regression equation of each index. Furthermore, based on the specification requirements of SBS modified asphalt as the index limit, the nonlinear equation system was derived using MATLAB (version 2021, MathWorks, Natick, MA, USA), and the optimal compound content of the ER and SBS modifier was analyzed according to the solution set. The three index tests and cloth viscosity tests of ER/SBS modified bitumen with the optimal compound content were performed and the simulation results were compared.

#### 2.3.3. Temperature Sweep Test

An advanced rotary rheometer (Anton Paar MCR 302 DSR instrument, Kegu Technology Development (Shanghai) Co., Ltd., Shanghai, China) was used to perform temperature sweep tests on ER/SBS composite modified bitumen before and after diesel corrosion to characterize the changes in the rheological properties of modified bitumen. The test adopts the strain control mode. The diameter of the parallel plate is 25 mm, and the thickness of the bitumen sample is 1.0 mm. The test temperature was set to 46 °C–82 °C, and the rotation frequency was 10 rad/s [24].

#### 2.3.4. Multiple Stress Creep Test

Multiple stress creep recovery (MSCR) was performed on a dynamic shear rheometer to test the modified bitumen before and after oil corrosion. The test temperature was set to 58 °C, 64 °C, and 70 °C, with stresses of 0.1 and 3.2 KPa. A total of 20 testing cycles were performed, and each cycle was of the form “creep (1 s)-recovery (9 s)” [25].

#### 2.3.5. BBR Test

The BBR (TE-BBR-F TYPE, CANNON, Shanghai Panan Testing Engineering Co., Ltd., Shanghai, China) test was used to evaluate the low-temperature bending creep performance of ER/SBS composite modified bitumen at −18 °C and −24 °C. According to the load and deformation values within 60 s, the creep stiffness (S) and creep rate (m) of ER/SBS composite modified bitumen was calculated. In addition, based on the calculated creep stiffness (S) and creep rate (m) data, the low-temperature crack resistance of the modified bitumen was evaluated [26].

#### 2.3.6. Fluorescence Microscopic Test

Fluorescence microscopy employs a point light source with high luminous efficiency that emits a certain light wavelength (e.g., ultraviolet light) as excitation light through the color filtering system. The fluorescent substances in the excitation specimen emit a variety of different colors of fluorescence, which are observed by the magnification of the objective lens and eyepiece. In this paper, a high-resolution fluorescence microscope (FM) (model: Axio Imager A2 m, manufacturer: CarlZeiss, Shenzhen Century Vision Electronic Equipment Co., Ltd., Shenzhen, China) was used to perform 10 × 10-fold fluorescence microscopic imaging of different types of asphalt before and after oil corrosion. The microscopic imaging changes of different modified asphalt and modified asphalt after oil corrosion were then analyzed [27].

## 3. Results and Discussion

### 3.1. Combined Design of Composite Modified Asphalt Content Based on the Uniform Experimental Method

Distinct to the traditional orthogonal experimental approach, the uniform design method abandons the “neat comparability” of the orthogonal experimental method, reduces the number of repetitions at the same level, and retains a more representative test point. Through the uniform design of the regression equation, the influence of each factor on the objective function is analyzed.

Cubuk et al. determined that when 2% epoxy resin was added to the matrix bitumen to modify the bitumen, the viscosity, softening point, and stability increased, while the heat sensitivity, surface energy, permeability, and peelability decreased [28]. Therefore, the X1 ER dosage used here for each of the 6 factor levels was 0.5%, 1%, 1.5%, 2%, 2.5%, 3%, while the X2 SBS modifier dosage used is 2.5%, 3%, 3.5%, 4%, 4%, 5%. The selection of each factor parameter and uniform design table are shown in Table 4, Table 5 and Table 6, respectively.

Each uniform design table corresponds to a usage table. If there are two factors, the first and third columns are selected for the experiment; if there are three factors, the first, second, and third columns are selected for the test; if there are four factors, the first, second, third, and fourth columns are selected for the trial. The last column (D) of Table 6 shows the deviation of the marking uniformity; the smaller the value of D, the better the uniformity. Combined with the ER and SBS modifiers in this paper, the first and third columns were selected for the experimental design. Table 7 integrates the parameters in Table 4, Table 5 and Table 6 to present the mixing combination of the two modifiers in this test. 

### 3.2. Analysis of Basic Physical Properties of Epoxy Resin/SBS Composite Modified Asphalt

According to the Test Regulations for Asphalt and Asphalt Mixtures for Highway Engineering (JTG E20—2011) [23], the penetration degree (25 °C), softening point (R&B), ductility (5 °C), and cloth viscosity (135 °C) of six dosage combinations of modified asphalt were tested, respectively (Table 8).

Figure 2 presents experimental results of the penetration degree of composite modified asphalt at 25 °C with the six mixing ratios. The ER content and penetration degree are linearly fitted, and the fitting determination coefficient of R^2^ = 0.92 indicates that there is a significant linear relationship between ER content and penetration degree. The penetration degree can indirectly reflect the isothermal viscosity characteristics of asphalt materials to a certain extent, and with the increase of ER content, the penetration value decreases regularly. Modified asphalts 2–5 meet the SBS modified asphalt penetration index. The results show that the addition of the modifier can effectively improve the viscosity of asphalt, significantly improve the high temperature performance of asphalt, and reduce the temperature sensitivity of asphalt.

Low-temperature ductility can characterize the low temperature performance of asphalt. Figure 3 depicts the effect of different amounts of ER on the ductility of asphalt at 5 °C. The ER content is negatively correlated with the 5 °C ductility ring value, which indicates that the incorporation of ER reduces the low temperature performance of asphalt to a certain extent. The proportion of SBS dosage is the percentage of SBS dosage to the total amount of modifier (i.e., the sum of SBS and ER contents in each dosage combination). Figure 4 shows the effect of different SBS dosage proportions on the ductility of asphalt at 5 °C. The proportion of SBS content is significantly linearly correlated with the ductility at 5 °C, and its fitting determination coefficient is 0.92. This indicates that the larger the proportion of SBS modifier content to the total amount of modifier, the higher the ductility value of composite modified asphalt and the better the low temperature performance of composite modified asphalt. Table 9 reveals that the ductility values of composite modified asphalt Nos. 1–4 are 32.2 cm, 30.6 cm, 20.2 cm, and 22.5 cm, respectively, which meet the requirements of the SBS modified asphalt ductility index (≥20 cm). The ductility values of composite modified asphalt Nos. 5–6 are 15.6 cm and 18.9 cm, respectively, which are lower than 20 cm. The ER content of asphalt Nos. 1–2 increased by 0.5% and 1%, respectively, and the SBS content was 3.5% and 5%, respectively (an increase of 1.5%). However, the ductility was reduced by 1.4 cm. The ER content of ER asphalt Nos. 2–3 increased by just 0.5% to 1% and 1.5%, and the SBS content decreased by 2% to 5% and 3%, respectively, while the ductility was reduced by 10.4 cm. According to the literature, SBS can improve the ductility of base asphalt [29]. Compared with No. 3 asphalt, the ER and SBS contents of No. 6 asphalt increased, but its ductility value still maintained a downward trend, which indicates that the ductility of asphalt is negatively affected by ER.

Figure 5 presents the total dosage of different dosage modifiers on the softening point. The fitting decision coefficient of 0.87 is significantly linear. Table 8 shows that the softening point of modified asphalt is 59.8 °C, which does not meet the qualified index of SBS modified asphalt softening point (≥60.0 °C). The softening point of asphalt Nos. 2–6 (75.6, 68.9, 76.2, 71.8, and 82.4 °C, respectively) all meet the qualified index. At the highest total modifier amount (7), its softening point exceeds 37% of the SBS qualified standard for modified bitumen. The total amount of modifier No. 2 and No. 6 is 6% and 7%, respectively, with a difference of only 1%, but the difference of softening point is 6.8 °C, which is less than that of No. 6 (3%) compared with ER content of No. 2 (1%). The total amount of modified asphalt for modifiers No. 2 and No. 6 was 6% and 7%, respectively, and the total amount of modifiers differed by only 1%, yet the difference value of the softening point was 6.8 °C, which is less than No. 6 (3%) compared with ER content No. 2 (1%). SBS modifier dosage No. 2 (5%) is greater than that of No. 6 (4%). The results show that ER plays a leading role in improving the high temperature performance of modified asphalt.

Figure 6 shows the effect of the total dosage of modifiers with different dosages on the viscosity of cloth at 135 °C. The fitting determination coefficient of R^2^ = 0.74 indicates that the viscosity of cloth has a strong linear relationship with the total modifier dosage. Table 8 report that modified asphalt Nos. 1, 2, 3, and 5 all meet the SBS modified asphalt qualification standards (≤3 Pa·s). The total dosage of modifiers was, from largest to smallest, No. 6 ≥ No. 4 ≥ No. 2 ≥ No. 5 ≥ No. 3 ≥ No. 1. The Brookfield viscosity at 135 °C exhibited the following trend (from largest to smallest): No. 6 ≥ No. 4 ≥ No. 5 ≥ No. 2 ≥ No. 3 ≥ No. 1. Combined with the fitting curve, the viscosity of asphalt is observed to be affected by the total content of these two modifiers, and the higher the total dosage of the modifier, the greater the viscosity of the asphalt.

### 3.3. Optimal Dosage Range Solution

#### 3.3.1. Determination of the Regression Model

We tested the composite modified asphalt with a uniform design combination of two factors and six levels. The four basic performance indicators of penetration degree, ductility, softening point, and cloth viscosity of each composite modified asphalt were determined via experiments. The ER dosage and SBS modifier content were selected as the basic performance influence indicators, and a mathematical model was established. A mathematical equation for the ER and SBS modifier dosages was then developed for the aforementioned four basic performance indicators using SPSS. The inequality equation system was established through the specification requirements of the SBS modified asphalt performance index and was solved via MATLAB to obtain the formula solution set that meets the qualified indicators of composite modified asphalt. The optimal composite dosage of modifiers was then derived through specific analysis. The softening point, penetration degree, cloth viscosity, and ductility indexes of SBS modified asphalt were selected as the limit values of the calculation model (Table 9).

The solution of the optimal compound dosage of composite modified asphalt is the superposition of the effect of a single modifier and the effect of a composite modifier. For epoxy resin dosage factor level X1 and SBS modifier dosage factor level X2, the functional regression model is described in Equation (1):(1)$$Y=K_{1}X_{1}+K_{2}X_{2}+K_{3}X_{1}X_{2}+K_{4}$$
where $K_{1}$, $K_{2}$, $K_{3}$, and $K_{4}$ are constants, $X_{1}$ and $X_{2}$ are the epoxy resin and SBS modifier indexes, respectively (%), and Y denotes the corresponding asphalt performance indicators.

The system of nonlinear inequalities for each index is shown in Equation (2):(2)$$\left\{ \begin{array}{l} -10.507X_{1}+0.605X_{2}-0.701X_{1}X_{2}+71.031\leq 60\\ -10.507X_{1}+0.605X_{2}-0.701X_{1}X_{2}+71.031\leq 40\\ 9.793X_{1}+5.877X_{2}-0.558X_{1}X_{2}+34.938\geq 60\\ -7.818X_{1}+1.787X_{2}+0.569X_{1}X_{2}+26.660\geq 20\\ 0.557X_{1}+0.285X_{2}+0.182X_{1}X_{2}-1.077\leq 3 \end{array}\right.$$

The set of inequalities given in Equation (2) were solved using MATLAB. Curves independent of the solution set are removed. Figure 7 plots the solution set graph. 

During the calculation process, no objective function is determined for the equation system, and thus the solution of the equation system is a solution set in the form of an array. Based on the literature and the results of previous experiments, the content ranges of SBS modified asphalt and epoxy resin are limited to 2.5–5% and 1–3%, respectively [28,30]. The results of the testing of composite modified asphalt No. 4 reveal that when the ER content is 2%, the output of SBS modifier is 4.5%, the Brookfield viscosity test result (3.32 Pa·s) is slightly higher than the index requirements. However, the rest of the indicators meet the requirements, and exhibit a good penetration degree and softening point performance. Therefore, the optimal dosage formula of composite modified asphalt should be similar to the compound dosage of modified asphalt No. 4. Based on the content ranges detailed above, an intersection point in the solution set of Figure 7 (2.3789, 3.8174) was selected. Considering the actual operating situation, the epoxy resin and SBS modifier contents were set as 2.3% and 3.8%, respectively.

#### 3.3.2. Optimal Dosage Performance Testing

In order to verify the performance of composite modified asphalt under the optimal dosage, the composite modified asphalt was prepared according to the ER and SBS modifier contents of 2.3% and 3.8%, respectively, and the penetration degree, ductility, softening point, and Brookfield viscosity of the composite modified bitumen were tested and compared (Table 10).

The predicted values in Table 10 were obtained by substituting the optimal dosage combination into Equation (2); errors of the predicted values of penetration degree, ductility, softening point, and 135 °C rotational viscosity index were determined as 3.7%, 10.7%, 2% and 1.7%, respectively. With the exception of the ductility error, all three indicators exhibited errors of less than 10%, indicating the reliability and practicability of the regression model established in this paper.

### 3.4. Effect of Oil Corrosion on Rheological Properties of Epoxy/SBS Composite Modified Bitumen

A traditional asphalt oil corrosion test mold is typically a stainless-steel mold. During the preparation of asphalt samples, petroleum jelly and other lubricants must be applied to the inner wall of the mold. Part of the lubricant will adhere to the demolded asphalt test piece, affecting the accuracy of the oil corrosion test results. In response to this, we employed a silicone mold to replace the stainless-steel mold due to its advantages of fast demolding, more accurate test results, low price, and so on.

Based on previous literature, the oil corrosion test applied diesel fuel to corrode the asphalt under 60 °C to model the most unfavorable high temperature situation in summer. A shear speed of 90 km/h was selected to simulate the driving speed of the vehicle based on the relationship between linear speed and angular velocity (6000 r/min shear head speed) under an oil immersion time of 10 min [31,32].The test mold and specimen are shown in Figure 8.

The complex shear moduli G* and δ were determined from the DSR test crucial for the characterization of the rheological properties of bitumen. The composite modified bitumen with different ER dosages was prepared by controlling the dosage of SBS modifier to 3.8%, with ER dosages of 1.3%, 1.8%, 2.3%, 2.8%, and 3.3%, respectively, and compared with the SBS modified asphalt content of 4.5%. The influence of different doses of ER on the rheological behavior of composite modified asphalt was investigated by temperature scanning (46–82 °C) to evaluate the high-temperature rutting resistance and fatigue cracking resistance of ER/SBS composite modified asphalt.

The complex shear modulus G* reflects the total ability of the asphalt to resist high temperature and deformation under the shear stress of periodic repetition. The greater the stiffness of the asphalt, the better the high temperature stability, and the stronger the ability to resist flow deformation. Figure 9 reveals that the G* values of both the matrix asphalt and modified asphalt decrease with the increase of temperatures within the range of 46–82 °C. This indicates that as the temperature increases, the asphalt sample is completely converted into a viscous fluid state, and the elastic components are completely transformed and disappear. This reduces the stiffness of the asphalt, as well as its ability to resist flow deformation, resulting in a decrease in G*. Compared with SBS modified bitumen without ER, the G* value of composite modified bitumen mixed with ER was observed to significantly increase. This is due to the formation of a strong cross-networked structure with the incorporation of ER, whereby the modified asphalt molecules are tightly wrapped. The composition of the modified asphalt is solidified and its ability to resist shear deformation at high temperatures is improved. The G* value of the composite modified asphalt with different ER contents is higher than that of SBS modified asphalt with content 4.5%. When the ER content increased from 1.8% to 2.3%, the G* value of composite modified asphalt increased significantly and then decreased as the ER content continued to increase. This is attributed to the increasing stability of the intermolecular asphalt structure with the ER content; the ability of asphalt to resist high-temperature shear will initially enhance and gradually becomes saturated.

Figure 10 presents the G* value of bitumen after diesel corrosion. The G* values of asphalt after oil corrosion were observed to decrease. The reduction rates of G* after the oil corrosion of 2.3% and 3.3% ER content composite modified asphalt at 60 °C were 36.9% and 35.5%, respectively, which is less than the G* reduction rate of 4.5% SBS modified asphalt (61.3%). This indicates that the oil corrosion resistance of composite modified asphalt doped with ER is stronger than that of 4.5% SBS modified asphalt. The G* value of composite modified asphalt with 2.3% ER content after oil corrosion is still slightly higher than that of 4.5% SBS modified asphalt without oil corrosion. This is because the oil in the bitumen can be soluble in organic solvents such as diesel, resulting in the destruction of the intermolecular structure of the asphalt, the softening of the asphalt, and the reduction of the deformation resistance of the asphalt. Moreover, the addition of ER absorbs oil, reducing the oil content of the asphalt, and locks part of the oil in the asphalt. At the same time, ER and SBS modifiers form a complex polymer network structure in the asphalt, wrapping the asphalt to play a “protective film role”. The diesel fuel has to break through this “protective film”, increasing the ability of asphalt to resist fuel corrosion.

Phase angle δ is a relative index that can evaluate the proportion of asphalt viscosity and elastic deformation. The material is considered an ideal absolute elastomer when δ = 0°; for δ = 90°, the material is considered to be an absolute viscous fluid. Figure 11 shows the phase angle change in the temperature range of 46–82 °C. The value of δ for different modified asphalt increases with the temperature. The composite modified asphalt doped with ER increases slowly at 70–82 °C compared with 4.5% SBS modified asphalt, with a significant plateau area due to the loss of the elastic component of asphalt from the higher heat with the increased temperature. The proportion of viscous components in asphalt gradually increases, and the elastic proportion is reduced. The asphalt thus gradually changes from an elastic to viscous form, resulting in an increase in δ. Moreover, under a certain temperature range, ER and SBS jointly build a network-like structure, which restricts the transformation of composite modified asphalt from a high elastic state to a viscous flow state. Thus, the growth of composite modified asphalt δ within 70–82 °C is slow. The largest to smallest phase angles of modified asphalt under different ERs were as follows; matrix bitumen, 4.5% SBS, 1.3% ER/3.8% SBS, 1.8% ER/3.8% SBS, 2.3% ER/3.8% SBS, 2.8% ER/3.8% SBS, and 3.3% ER/3.8% SBS. This indicates that the molecular motion chain of SBS modified asphalt was inhibited after the incorporation of ER, such that the elastic behavior of modified asphalt increased and the δ was reduced. Figure 12 presents the changes in δ after oil corrosion. Following the incorporation of ER, the change in δ after the oil corrosion of composite modified asphalt was not obvious compared with the 4.5% SBS modified asphalt. This is because the active components in the asphalt (e.g., oil and gum) are dissolved after oil corrosion, which changes the structure of the asphalt components, reducing the stiffness modulus of pure asphalt in the modified asphalt. This softens the asphalt, increases the proportion of viscous components, and δ becomes larger, thus weakening the high temperature deformation resistance of the asphalt. When diesel infiltrates into ER/SBS composite modified asphalt, ER and SBS form a cross-network structure in the asphalt which prevents diesel fuel from dissolving the active components in the asphalt. As a consequence, the δ effect of oil corrosion on the composite modified asphalt is not significant, and the composite modified asphalt can still maintain a good resistance to high temperature deformation after oil corrosion.

### 3.5. Analysis of Multiple Stress Creep Test Results

Previous studies have revealed that the penetration, ductility, and softening point are not able to characterize the rutting resistance performance index of asphalt under repeated vehicle loading [33]. In order to better evaluate the ability of asphalt to resist permanent deformation, the MSCR test is implemented with DSR. The average strain recovery rate (R) and unrecoverable creep flexibility (Jnr) are commonly used in MSCR tests to evaluate the test results. R reflects the elastic deformation recovery ability of asphalt specimens, the larger the R value of the asphalt binder, the better the elasticity, and the better the high temperature deformation resistance. JNR reflects the ability of asphalt slurry to resist permanent deformation, and the higher the value, the weaker the high temperature rutting resistance. The clearance between the 25 mm rotor and the parallel plate under the DSR fixture is 1 mm, and a complete test is divided into the two stages of loading and unloading. R and Jnr are calculated as in Equations (3) and (4):(3)$$R=0.1\sum_{i=1}^{10}\frac{\gamma_{ip}-\gamma_{inr}}{\gamma_{ip}-\gamma_{io}}$$
(4)$$J_{nr}=0.1\sum_{i=1}^{10}\frac{\gamma_{inr}-\gamma_{io}}{\tau }$$
where $\gamma_{p}$ is the peak strain for each cycle, $\gamma_{o}$ is the initial strain for each cycle, $\gamma_{nr}$ is residual strain per cycle, and τ is the loading stress. In this study, MSCR tests were performed on the original bitumen, 4.5% SBS modified bitumen, and ER/SBS composite modified bitumen with different ER contents.

From Figure 13, it can be seen that the R value gradually decreases from 0.1 kPa and 3.2 kPa as the stress changes. Under the same stress conditions, the R value also gradually decreases as the temperature increases. For high-stress conditions, an increase in temperature results in a lower R value, and when the stress changes significantly, the R value decreases significantly. This indicates that the high temperature and high stress will reduce the high-temperature elasticity of the asphalt, and with the increase of the ER content in SBS modified asphalt, the R value gradually becomes larger. When the ER content in ER/SBS composite modified asphalt increased from 1.8% to 2.3%, the R value increased significantly. However, this was not the case when the ER content increased from 1.8% to 3.3%, revealing that the addition of ER promoted the cross-linking of solid networks and improved the elastic properties of asphalt. In addition, the network structure formed by the ER molecules in the modified asphalt tended to be stable, resulting in an insignificant increase in the R value. Figure 14 shows the R values of asphalt after oil corrosion decreased under the stress conditions of 0.1 Kpa and 3.2 Kpa, while the R values of matrix asphalt, 4.5% SBS modified bitumen, 2.3% ER/SBS composite modified asphalt, and 3.3% ER/SBS composite modified asphalt decreased by 67.2%, 81%, 39.9%, and 38.2%, respectively. The R values of the two composite modified asphalts were observed to decrease more than those of SBS modified asphalt and matrix bitumen. Moreover, the R value of the composite modified asphalt after oil corrosion exceeded that of SBS modified asphalt without oil corrosion, indicating that the composite modified asphalt exhibits oil corrosion resistance at lower temperatures and stress.

Figure 15 reveals that the Jnr value gradually increases from 0.1 Kpa and 3.2 Kpa as the stress value changes. Under the same stress conditions, the Jnr value gradually increases with temperature, and the increased amplitude of Jnr under high stress is greater than that under low stress. This indicates that high temperature and stress will reduce the deformation resistance of asphalt. The results of Jnr0.1 and Jnr3.2 demonstrate that the sensitivity of composite modified asphalt doped with ER to stress and temperature is much lower than that of matrix asphalt and 4.5% SBS modified asphalt, and the Jnr3.2 value gradually decreases with ER content at 3.2 Kpa. When the ER content increases from 1.8% to 2.3% (2.3% to 3.3%), the Jnr3.2 value decreases significantly (non-significantly). The change trend of asphalt Jnr with temperature after oil corrosion is roughly the same as that of asphalt before oil corrosion (Figure 16). After oil corrosion, the Jnr value of the four types of asphalt increased more and more with the increase of temperature. This is attributed to the high sensitivity of asphalt to temperature after oil corrosion damage. In particular, the higher the temperature, the more intense the movement of residual diesel molecules inside the asphalt after oil corrosion, and the greater the loss of active ingredients in the asphalt. This damages the internal structure and weakens the high temperature resistance of asphalt. At the temperature and stress of 3.2 Kpa and 70 °C, respectively, the added value of the matrix asphalt and SBS modified asphalt Jnr is much greater than that of the composite modified asphalt with ER contents of 2.3% and 3.3%. Moreover, the added value of Jnr for composite modified asphalt with 2.3% (3.3%) ER content after oil corrosion is 2.25 (4.92). This is attributed to the fact that matrix asphalt and SBS modified asphalt have more light components; diesel molecules penetrate into the asphalt resulting in the thinning of the intermolecular adsorption layer inside the asphalt so that some light components cannot be fully adsorbed and loosened. ER’s own unique properties can adsorb the light components in the asphalt, forming a dense cross-network-like structure, preventing the further infiltration of diesel molecules; excess ER which does not participate in the construction of the network structure will be distributed in the asphalt molecules. Rapid heat conduction accelerates the movement of diesel molecules. This inhibits the improvement of the oil corrosion resistance of asphalt. The comprehensive analysis of the MSCR test results shows that when the ER content is 2.3%, the composite modified asphalt produces small shear deformation under the action of repeated loading and unloading, and the high temperature deformation resistance is the most significant. Furthermore, the smallest increase in Jnr is observed after oil corrosion, while the permanent deformation resistance is the strongest, which is generally consistent with the temperature sweep test results.

### 3.6. Low Temperature Cracking Performance of Epoxy Resin/SBS Composite Modified Asphalt

In order to characterize the effect of ER content on the creep performance at low temperatures, the creep performance of ER/SBS composite modified asphalt at −16 °C and −24 °C was tested by a bending beam rheometer. Stiffness modulus (S) and the creep rate (m) of 60.00 s were used to evaluate the results, where S indicates the ability of the asphalt to resist low temperature deformation, and m represents the stiffness modulus degree of the asphalt changes with creep time. The smaller the S value, the lower the risk of the cracking of the asphalt binder at low temperatures; the larger the m value, the stronger the stress relaxation ability of the asphalt, and the smaller the probability of the asphalt cracking failure. The AASHTO (American Association of State Highway and Transportation Officials) requirements state that creep stiffness S ≤ 300 Mpa or m ≥ 0.3. Figure 17 presents the trends of S and m for composite modified bitumen with ER content. 

When the temperature increases from −24 °C to −18 °C, the stiffness modulus (S) of the modified asphalt decreases significantly, while the creep rate (m) increases significantly. This indicates that the modified asphalt has a stronger low-temperature deformation resistance and low-temperature crack resistance at −18 °C compared to at −24 °C with each ER content. At −18 °C and −24 °C, the creep stiffness (S) value exhibits a gradual increases trend with the increasing ER content, and the fluctuation amplitude of the ER content change on m and S at −18 °C is less than that at −24 °C. This reveals that the incorporation of ER has no significant effect on the asphalt performance at this temperature. Under low temperature conditions, with the increase of ER content, the S value of modified asphalt exhibits a trend of becoming larger, and the m value exhibits a decreasing trend. When the ER content is greater than 2.3%, the m and S values change significantly (decreasing and increasing rapidly, respectively). At −24 °C, the m and S values of the modified asphalt were 0.24 and 308.4, respectively, under the ER content of 3.3%, which does not meet the specification requirements.

The results reveal that the appropriate amount of ER incorporation will not have a significant impact on the low-temperature fission and creep resistance and toughness of the modified asphalt and can meet the road requirements under low temperature conditions. An excessive ER incorporation may strongly affect the interaction between asphalt molecules, which greatly reduces the toughness, low-temperature crack resistance, and the stress relaxation ability of the modified asphalt. Combined with the DSR and BBR test data, when the ER content is 2.3%, the composite modified asphalt can also induce excellent high-temperature rheological properties, meeting the basic low-temperature rheological properties.

### 3.7. Fluorescence Microscopic Analysis

Fluorescence microscopy is the most commonly used technique for evaluating the dispersion state of modifiers in polymer-modified bitumen. In this paper, the matrix bitumen, 4.5% SBS modified bitumen, and 2.3% ER + 3.8% SBS composite modified bitumen before and after oil corrosion are tested (Figure 18).

The three-component method separates asphalt into oil, resin, and asphaltene, of which the oil content accounts for about 45–60% of asphalt. The fluorescence microscope observations of asphalt reveal that the aromatic content in asphaltene oil usually exhibits the strongest fluorescence signal, while asphaltene does not emit fluorescence. This is because the matrix asphalt is a single-phase matrix. The matrix asphalt observed by the fluorescence microscope is a homogeneous material (Figure 18a). The modifier in the modified bitumen absorbs the aromatic content of the bitumen, and the matrix bitumen and modifier show different colors under the irradiation of the fluorescent light source [34]. The bright color of the modifier in Figure 18c,e is caused by the wavelength difference of the reflected wave of the asphalt and the modifier to the fluorescent light source. Figure 18c shows that at the SBS content of 4.5%, SBS is distributed in the asphalt as large particles. When the ER incorporation is 2.3%, the modifier in the bitumen is more densely dispersed and evenly distributed in the asphalt in the form of small particles, forming a strong cross-networked structure (Figure 18e). This is due to the existence of the primary phase separation between ER and SBS modified bitumen and the secondary phase separation between bitumen and SBS. In addition, the phase separation in the ER/SBS composite modified bitumen destroys the original dispersion state of the SBS particles in the asphalt [35], resulting in the redistribution of SBS in smaller spherical particles. Compared with Figure 18a,b, which show the diesel and diesel corrosion to produce serious gully-like damage on the asphalt, Figure 18c,d reveal that the number of SBS modified asphalt modifiers after oil corrosion is greatly reduced, and still produces ravine-like damage. For the ER/SBS composite modified asphalt after oil corrosion, although the modifier particles are slightly reduced, there is no gully-like failure as with the oil-eroded asphalt (Figure 18e,f). This is because the oil in the asphalt is easily soluble in organic solvents such as diesel [4], and thus some of the asphalt dissolves with diesel, while the aromatic content of the reflected fluorescence is greatly reduced, and ravine-like damage is observed by the fluorescence microscope. With the incorporation of ER, a strong network structure is formed with ER, SBS and asphalt, which hinders the infiltration of diesel molecules. The incorporation of ER also locks part of the oil in the asphalt, reducing the loss of oil in the asphalt damaged by oil corrosion.

## 4. Conclusions

In this paper, ER/SBS composite modified asphalt was prepared by physical blending and the basic performance indicators of composite modified asphalt were experimentally analyzed by the uniform design method. The optimal dosage ratio of SBS and ER modifier in composite modified asphalt was then obtained and the composite modified asphalt with different ER amounts was subjected to pre- and post-oil corrosion DSR, pre- and post-oil corrosion fluorescence microscopy, and BBR tests. Following this, modified asphalt was compared with 4.5% SBS modified asphalt. The main research conclusions of this paper are as follows:(1)Through balancing improvement and cost, the optimal dosage ratio of ER/SBS composite modified asphalt is 2.3%/3.8%.(2)The results of the DSR temperature sweep test show that with the increase of ER content, the G* of SBS modified asphalt increases, the δ decreases, the G* of modified asphalt decreases, and the δ increases after oil corrosion. Moreover, the reduction rate of composite modified asphalt G* after oil corrosion is less than that of SBS modified asphalt, indicating that the incorporation of ER reduces the temperature sensitivity of SBS modified asphalt, enhances the resistance to deformation at high temperatures, and improves the resistance of asphalt to fuel corrosion damage.(3)The results of the MSCR tests showed that the incorporation of ER can improve the elastic recovery ability of the SBS modified asphalt, while the R (Jnr) value of modified asphalt decreased (increased) after oil corrosion. This indicates that diesel can weaken the elastic recovery ability of asphalt. The performance improvement effect is most significant at the ER content of 2.3%, while the R value reduction rate and Jnr value increase rate are the smallest following the damage to the modified asphalt oil corrosion, and the ability to resist diesel corrosion is the strongest.(4)The results of low-temperature cracking performance tests show that ER incorporation has a negative impact on the low-temperature performance of SBS modified asphalt. In particular, when the ER content does not exceed 2.3%, the low-temperature performance of asphalt decreases slowly, while for ER content greater than 2.3%, the low-temperature performance of asphalt deteriorates significantly. The creep rate m is less than 0.3 and the S value is greater than 300, which does not meet the specification requirements.(5)The microscopic morphology of the three types of asphalt reveals that the SBS modifiers are loosely distributed in the asphalt. With the incorporation of ER, the phase separation in ER/SBS composite modified asphalt destroys the original dispersion state of SBS particles in asphalt, and the modifiers are densely and uniformly distributed in the matrix asphalt. After the oil corrosion, the three asphalt types showed different degrees of damage, among which the matrix asphalt exhibited the most serious damage, followed by SBS modified asphalt. ER/SBS composite modified bitumen had the smallest degree of damage, indicating that it could effectively resist oil corrosion damage.

Epoxy resin has excellent fuel corrosion resistance, and due to price factors the previous scholars’ research on epoxy resin is mostly bridge deck paving, but in road engineering it is rarely studied; this paper proposes the application of low content epoxy resin in road engineering. Through indoor experiments, it was found that when the ER content is 2.3%, SBS modifier content is 3.8%, composite modified asphalt shows strong pavement performance. The results of the current study will encourage further research into the production of modified pitch mixtures using low-content ER modifiers. This work also has promotional significance for the practical application of ER/SBS high-performance modified asphalt.

## Figures and Tables

**Figure 1 polymers-15-00104-f001:**
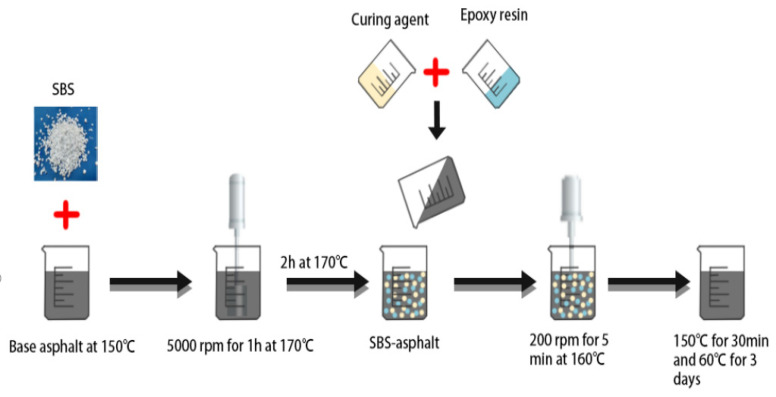
Flow chart of the preparation of ER/SBS composite modified asphalt.

**Figure 2 polymers-15-00104-f002:**
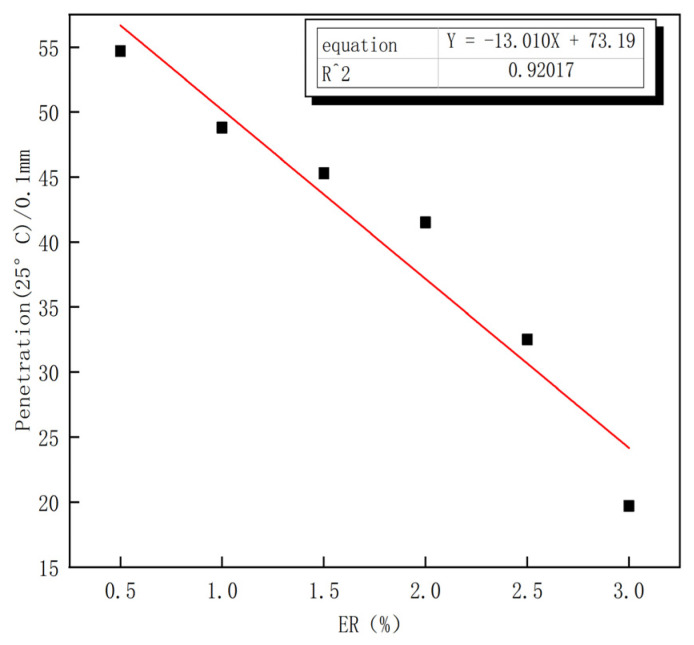
Relationship between ER content and penetration degree.

**Figure 3 polymers-15-00104-f003:**
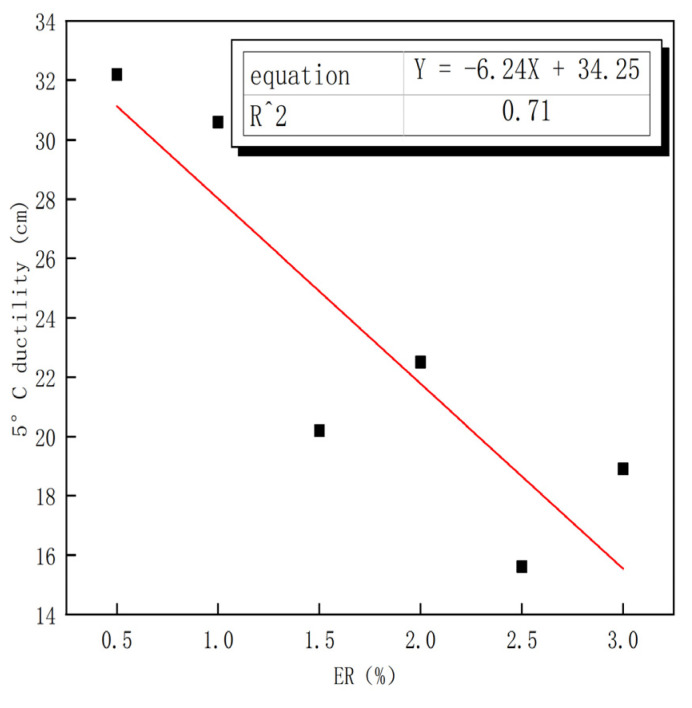
Relationship between ER content and low temperature ductility.

**Figure 4 polymers-15-00104-f004:**
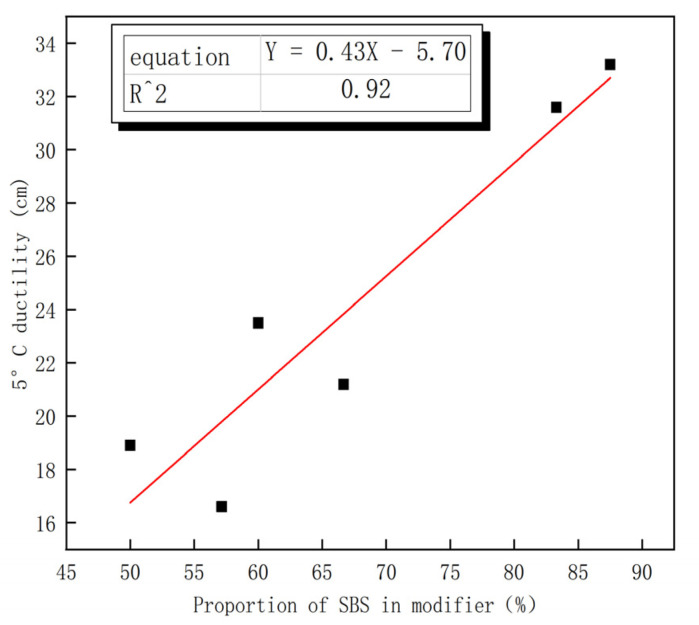
Relationship between Proportion of SBS in modifier and low temperature ductility.

**Figure 5 polymers-15-00104-f005:**
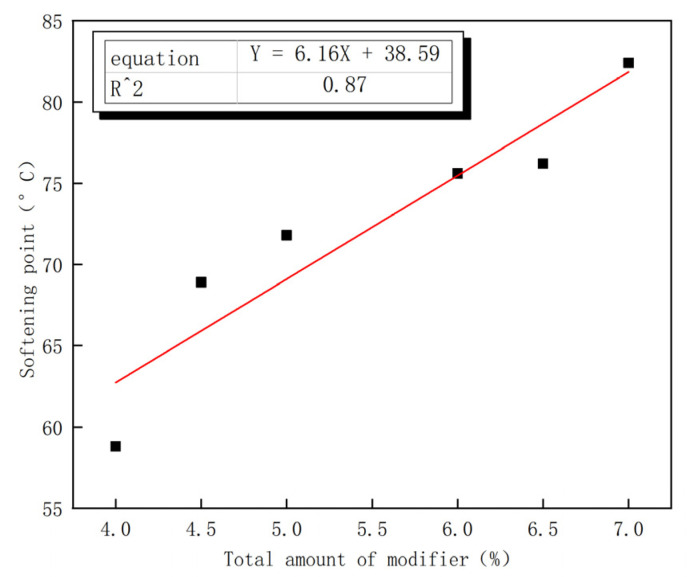
Relationship between total modifier and softening Point.

**Figure 6 polymers-15-00104-f006:**
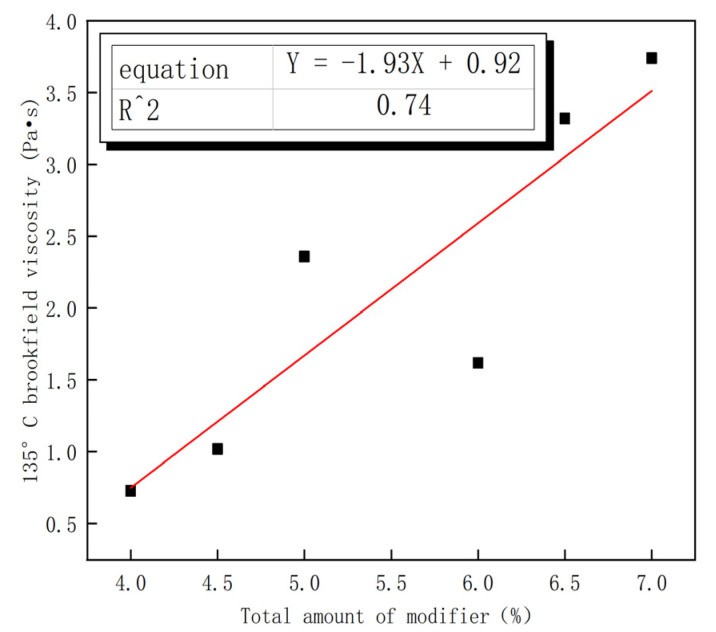
Relationship between total modifier content and brookfield viscosity.

**Figure 7 polymers-15-00104-f007:**
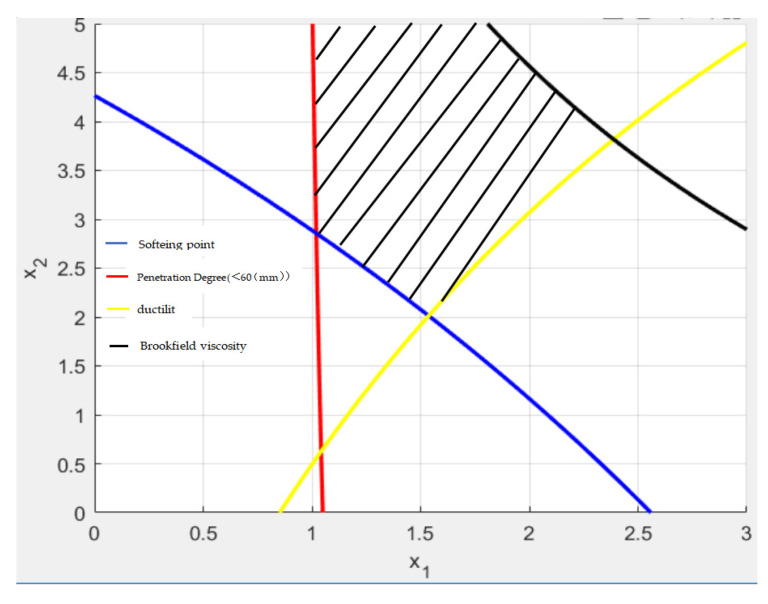
The best composite dosage of modifier in the disaggregation region.

**Figure 8 polymers-15-00104-f008:**
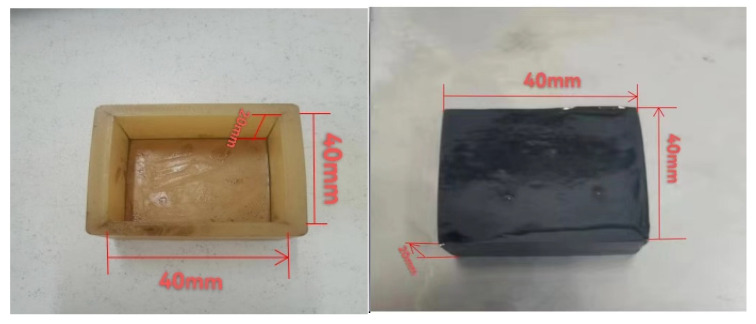
Self-made silicone mold and demolding asphalt test piece.

**Figure 9 polymers-15-00104-f009:**
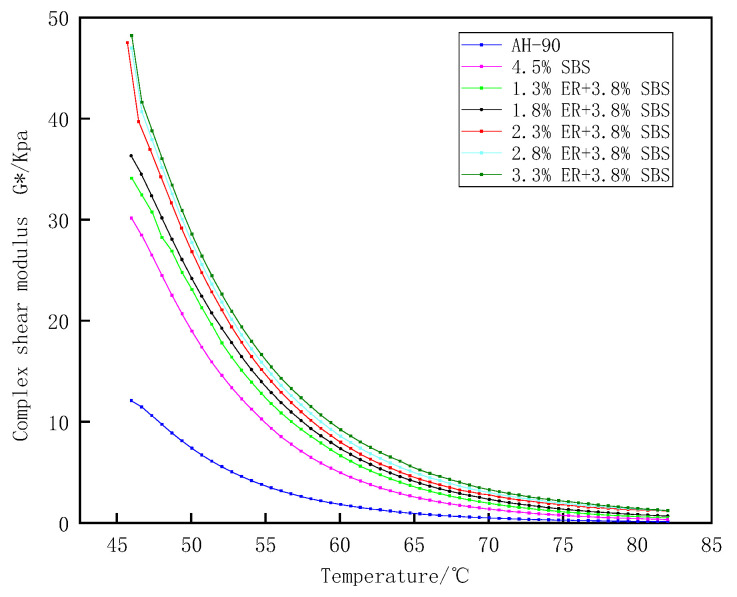
Effect of temperature on the shear modulus of ER/SBS modified bitumen.

**Figure 10 polymers-15-00104-f010:**
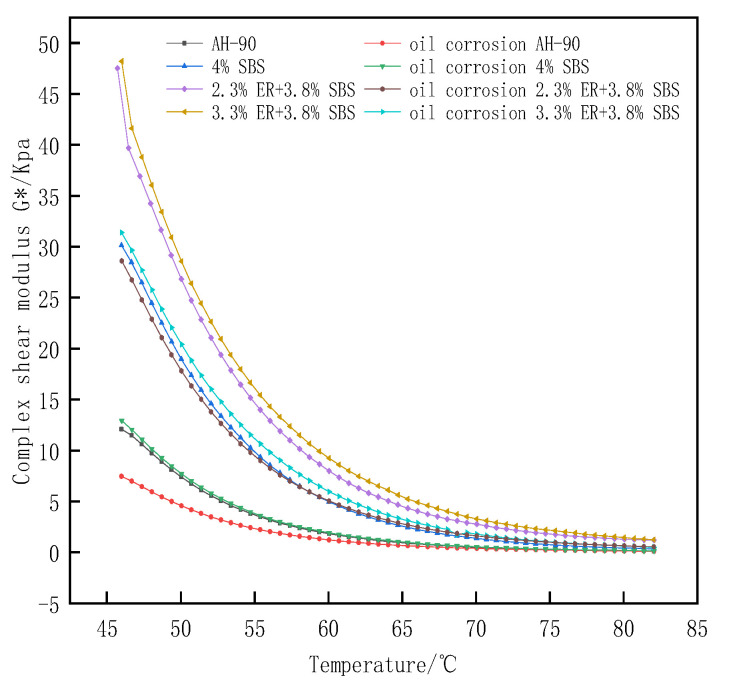
Effect of oil corrosion on the composite shear modulus of ER/SBS modified bitumen.

**Figure 11 polymers-15-00104-f011:**
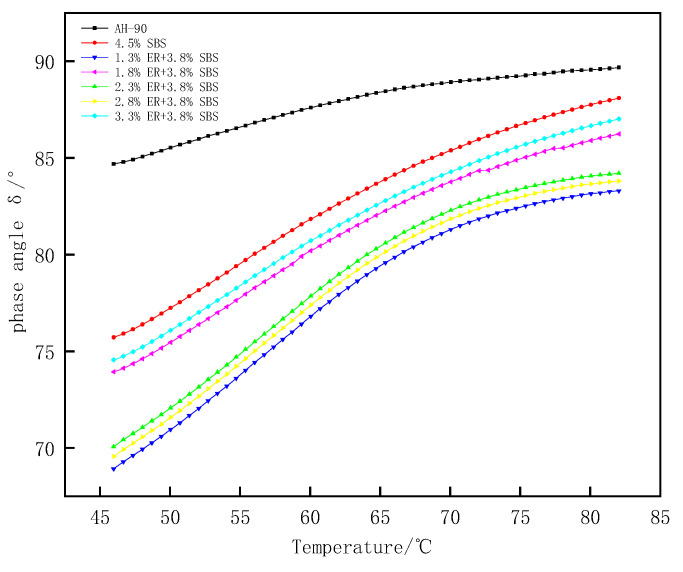
Effect of temperature on the complex phase angle of ER/SBS modified bitumen.

**Figure 12 polymers-15-00104-f012:**
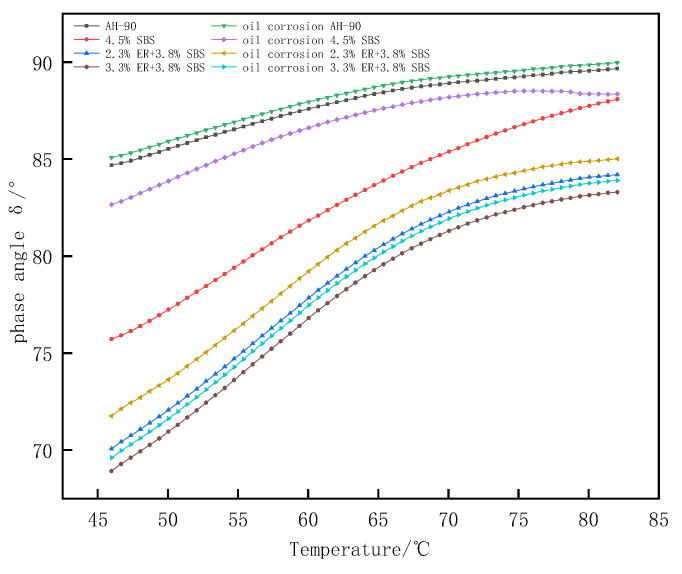
Effect of oil corrosion on the phase angle of ER/SBS modified bitumen.

**Figure 13 polymers-15-00104-f013:**
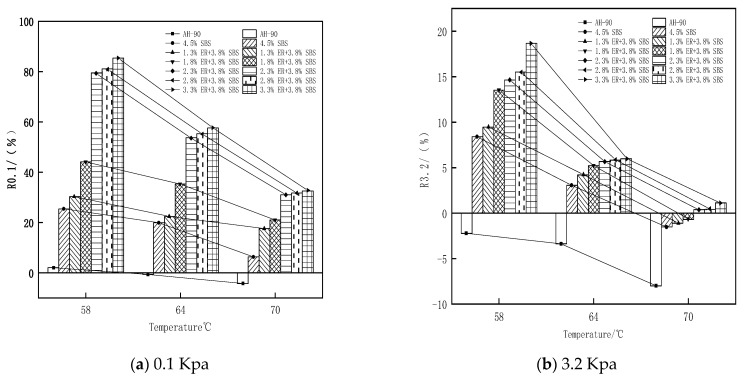
Average strain creep recovery rate of each bitumen with temperature under different stresses.

**Figure 14 polymers-15-00104-f014:**
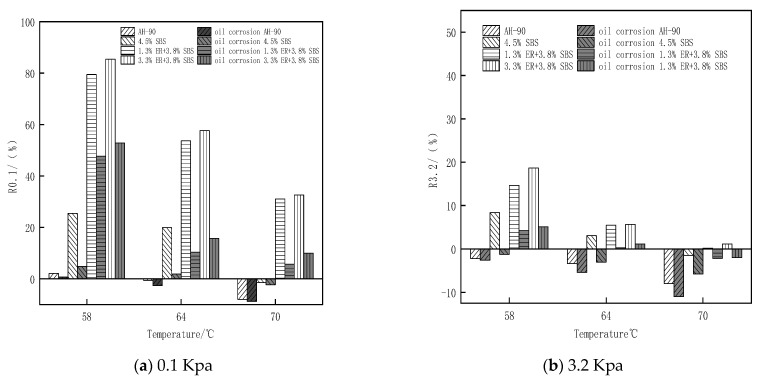
Changes in the average strain creep recovery rate of each bitumen after oil corrosion.

**Figure 15 polymers-15-00104-f015:**
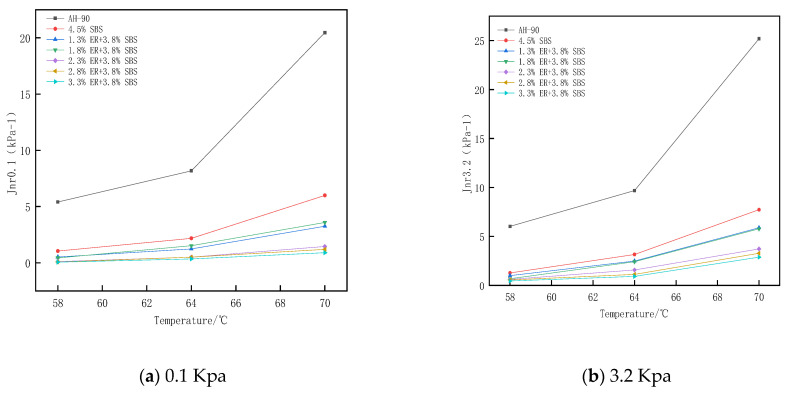
Changes in the irrecoverable creep flexibility of each bitumen with temperature under different stresses.

**Figure 16 polymers-15-00104-f016:**
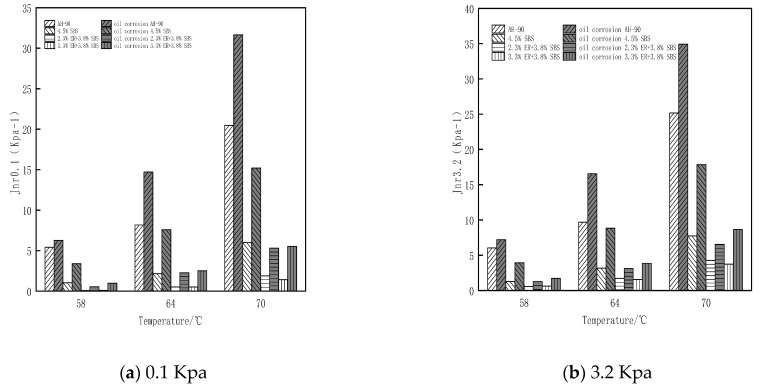
Changes in the flexible amount of irrecoverable creep of each bitumen after oil corrosion.

**Figure 17 polymers-15-00104-f017:**
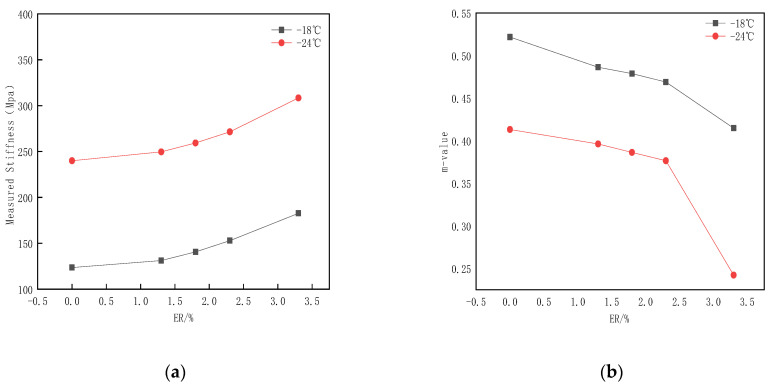
Effect of ER content on the low temperature performance of SBS modified bitumen: (**a**) Effect of ER content on the stiffness modulus; (**b**) Effect of ER content on the creep rate.

**Figure 18 polymers-15-00104-f018:**
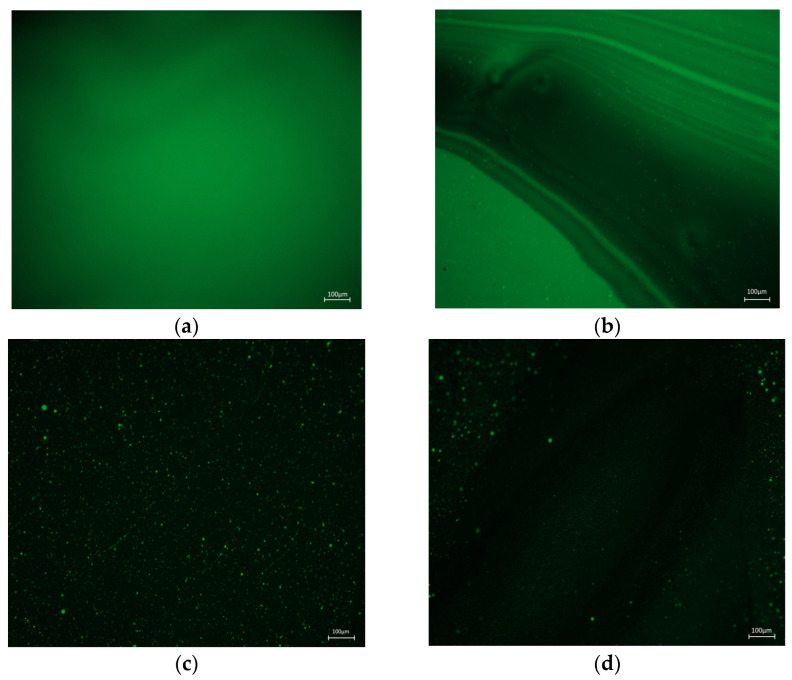
Microscopic images of three kinds of asphalt fluorescence before and after oil corrosion: (**a**) Matrix asphalt fluorescence microscopy; (**b**) Oil corrosion matrix asphalt fluorescence microscopy; (**c**) 4.5% SBS modified asphalt fluorescence microscope; (**d**) Oil corrosion 4.5% SBS modified asphalt fluorescence microscope; (**e**) 2.3% ER/3.8% SBS fluorescence microscopy; (**f**) Oil corrosion 2.3% ER/3.8% SBS fluorescence microscope.

**Table 1 polymers-15-00104-t001:** Technical indicators of SK-90 matrix asphalt.

Tested Variable	Technical Requirements
Penetration degree (100 g, 5 s, 25 °C)/(0.1 mm)	80~100
Softening point/°C	≥45
Ductility (50 mm/min, 5 °C)/cm	≥25
Solubility (%)	≥99.9
Dynamic viscosity 60 °C (Pa·s)	≥160

**Table 2 polymers-15-00104-t002:** Technical indicators of SBSYH-792E.

Tested Variable	Technical Requirements
Tensile strength/(Mpa)	≥24
Hardness shore/(A)	≥85
MFR (g/10 min)	0.1~5.0
25% toluene solution viscosity/(Mpa·s)	850~1850

**Table 3 polymers-15-00104-t003:** Technical indicators of E-44 epoxy resin.

Tested Variable	Technical Requirements
Epoxy/(g/eg)	210~240
Softening point/°C	12~20
Hydrolysable chlorine/(wt%)	≤0.3
Volatile/(wt%)	≤0.6

**Table 4 polymers-15-00104-t004:** Factor levels.

Level	Factor	
	X1-Epoxy Resin Content (%)	X2-SBS Content (%)
1	0.5	2.5
2	1	3
3	1.5	3.5
4	2	4
5	2.5	4.5
6	3	5

**Table 5 polymers-15-00104-t005:** U6(64) Uniform design table.

Serial Number	1	2	3	4
1	1	2	3	6
2	2	4	6	5
3	3	6	2	4
4	4	1	5	3
5	5	3	1	2
6	6	5	4	1

**Table 6 polymers-15-00104-t006:** U6(64) uses tables.

Number of Factors	Column Number				D
2	1	3	—	—	0.1875
3	1	2	3	—	0.2656
4	1	2	3	4	0.2990

**Table 7 polymers-15-00104-t007:** Modifier dosage combination.

Dosage Combination	X1—Epoxy Resin Content (%)	X2—SBS Content (%)
1	0.5	3.5
2	1	5.0
3	1.5	3.0
4	2	4.5
5	2.5	2.5
6	3	4.0

**Table 8 polymers-15-00104-t008:** Test results of basic physical properties of composite modified asphalt.

Variable	1	2	3	4	5	6	SBS Modified Asphalt Index
Penetration (25 °C)/0.1 mm	64.7	58.8	55.3	51.5	42.5	29.7	40~60
5 °C ductility (cm)	32.2	30.6	20.2	22.5	15.6	18.9	≥20
Softening point/°C	59.8	75.6	68.9	76.2	71.8	82.4	≥60.0
135 °C Brookfield viscosity (Pa·s)	0.73	1.62	1.02	3.32	2.36	3.74	≤3

**Table 9 polymers-15-00104-t009:** Performance index of SBS modified asphalt.

Index	Specification
Penetration degree (25 °C)/0.1 mm	40–60
Softening point/°C	≥60.0
135 °C Brookfield viscosity /(Pa·s)	≤3
5 °C ductility (cm)	≥20

**Table 10 polymers-15-00104-t010:** Optimal dosage-asphalt performance test results.

Project	Predicted Value	Title 3	Error
Penetration degree (25 °C)/0.1 mm	43.0	44.6	3.7%
Ductility (5 °C)	20.5	22.7	10.7%
Softening point/°C	74.7	76.2	2%
135 °C rotational viscosity/(Pa·s)	2.88	2.83	1.7%

## Data Availability

The data presented in this study are available on request from the corresponding author.

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
