# Peer review of "A Study on High and Low Temperature Rheological Properties and Oil Corrosion Resistance of Epoxy Resin/SBS Composite Modified Bitumen"

_polymers, 2022, doi:10.3390/polym15010104_

Round 1

Reviewer 1 Report

This is a fine done-written paper based upon good laboratory research. Experiments are well planned, and the analyses were affected by appropriate methods. There is a sufficient discussion of the results obtained. I recommend accepting the article for publication in the Polymers journal after some corrections:

1. Missing information about the authors (institute, city, country)

2. The abstract is very long. Need to cut

3. Line 83: "the poor compatibility between ER and asphalt". This statement should be confirmed by a reference (e.g. 10.1002/pen.25399, where the miscibility of epoxy resin with an asphaltene/resin blend was studied).

4. Line 98-112: This is information for authors. She needs to be removed

5. Give a description of epoxy resin and sbs. Photos of these components are not needed. They don't provide any information

6. Figure 19 needs a scale

7. Write conclusions

Reviewer 2 Report

IN this work, I have found no interesting points nor any novelty from from the authors.

Especially, this manuscript is not suitablt to be published in pomymers journal. this work includes only typical data for ER/SBS composites modified asphalt. 

Too many figures that are not actually required to consider. abstarct is too lengthy and does not convey the real essence of the work.

Therefore, I do not recommend this work to be published in polymers journal. 

Reviewer 3 Report

The paper entitled 'Study on high and low temperature rheological properties and oil corrosion resistance of epoxy resin/SBS composite modified bitumen' could be of interest to other researchers and manufacturers of the materials used in the study. However, a number of shortcomings and errors were noted in the paper, particularly editing or ambiguity. It gives the impression that the authors were in a great hurry when preparing the manuscript, or were doing it for the first time, which unfortunately makes the paper lose its value.

Below are my comments, both editorial and substantive:

1. Line 6 - Lack of affiliation 

2. I find the literature review in the introduction insufficient. Papers of similar scope have been published on the topic under discussion, I suggest reviewing them. 

3. Lines 98-112, I have the impression that this is text left out of the formatting. Such editing errors are unacceptable. 

4. Line 117 - no reference to the literature where the information on the materials used was taken from

5. Table 1 - please check spaces and spacing and units for individual properties.

6. Line 119 - Figure 1 contributes nothing, from my perspective it is not needed in the paper. 

7. The paper lacks an explanation as to why the materials were used. Could a different resin be used? 

8. I don't understand table 3, what is shown in it? The text lacks references to tables. 

9. Figure 9 also lacks substantive meaning. A drawing indicating the dimensions of the samples made would be better.

10. The "Discussion of results" section should include references and comparisons to other studies carried out to date in the area under discussion. This part of the manuscript should be expanded. 

Round 2

Reviewer 2 Report

I still found no novelty nor any other improved points from this article.

Reviewer 3 Report

The revised manuscript takes into account my earlier comments. 

In my opinion, it can be published in the Polymers journal